# Research on the Modeling of Automatic Pricing and Replenishment Strategies for Perishable Goods with Time-Varying Deterioration Rates

**Aihua Gu \*, Zhongzhen Yan, Xixi Zhang and Yongsheng Xiang**

School of Information Science and Technology, Yancheng Teachers University, Yancheng 224002, China; yenny2023@163.com (Z.Y.); 19895972270@163.com (X.Z.); 19850971955@163.com (Y.X.)
\* Correspondence: guaihua1978@163.com

**Abstract:** This paper focuses on the modeling of automatic pricing and replenishment strategies for perishable products with time-varying deterioration rates based on an improved SVR-LSTM-ARIMA hybrid model. This research aims to support supermarkets in planning future strategies, optimizing category structure, reducing loss rates, and improving profit margins and service quality. Specifically, the paper selects perishable vegetables as the research category and calculates the cost-plus ratio for each vegetable category. Correlation analysis is conducted with total sales, and a non-parametric relationship curve is obtained using support vector regression for nonlinear fitting. The long and short memory recurrent neural network is then used to predict sales volume, and a pricing strategy is calculated based on the fitting curve. Additionally, the paper establishes a correlation between loss rate and shelf life, corrects the daily average sales volume index, and solves the problem of quantity and category of replenishment using a backpack problem approach. By considering multiple constraints, a quantitative category replenishment volume and pricing strategy is obtained. The mathematical model proposed in this paper addresses the replenishment and pricing challenges faced by supermarkets, aiming to improve revenue and reduce loss while meeting market requirements.

**Keywords:** data mining and analysis; support vector machines; neural networks; supermarket sales strategy; cost-plus pricing

**MSC:** 62P20; 68T07




## 1. Introduction

Perishable goods lose their economic value due to gradual physical deterioration such as decay and evaporation, which results in changes in quality and quantity over time. In addition, the deterioration of these products generates waste, the disposal of which incurs new costs [1]. The perishable goods market is highly competitive, as perishable products, which account for a large portion of supermarket sales, are a means for retailers to differentiate themselves from competitors. The quality and variety of perishable foods have become critical factors for winning orders from consumers, who now have high standards for the quality and freshness of products. To ensure the quality of perishables, many merchants have set up automated pricing and replenishment strategies for perishables. Taking into consideration the impact of the quantity and price of goods on the production and operation process of retailers, it is thus necessary to study the automatic pricing and replenishment strategies of inventory models for perishable products.

In 1963, Ghare and Schrsder studied the inventory model of perishable products and observed that certain products would undergo partial deterioration over time, which could be appropriately characterized as a negative exponential function of time. This discovery led them to propose the EOQ model for the first time, which laid the foundation for subsequent research on perishable product inventory [2]. After conducting researches, Ghare

and Schrade found a negative exponential relationship between time and the decline in the inventory levels of perishable goods. They established an exponential decay inventory model that suited continuous deterioration at a constant deterioration rate. This model provides a basis for the calculation of real-time and instantaneous inventory levels in many studies on perishable products.

In supermarkets, the quantity of goods can have a certain impact on demand, which leads to price changes. Levind [3] and Silver [4] elucidated this relationship, where the sales volume of certain products is positively correlated with the number of products on the shelf. The more products on the shelf, the greater the room for the consumer's choice and the higher the sales volume.

Harris proposed the classic EOQ model, which assumes that inventory items have an infinite shelf life and can be stored indefinitely to meet future demand. However, certain types of goods may deteriorate or expire over time, making them unstable [5].

Nahmias provided the first comprehensive review on how to determine appropriate ordering strategies for a perishable inventory with fixed usage periods and an inventory that follows continuous exponential decay [6]. Based on this discovery, researchers began analyzing strategies for perishable inventory, where demand depends on the inventory levels of display areas. Mandalay and Phaujar considered a production inventory model for perishable goods where demand is linearly dependent on inventory levels and productivity is determined [7]. Raafat presented a survey of the literature up until 1990 on inventory models featuring continuous deterioration [8]. Goyal and Giri further extended the work of Nahmias and Raafat, providing an overview of product degradation and deterministic and stochastic demands [9]. Duan studied the question of the inventory optimization of perishable goods with demand dependent on inventory [10]. Bakker continued the work of Goyal and Giri, organizing and summarizing 227 relevant studies from 2001 to 2011 [11]. In 2016, Janssen used the same classification method based on Bakker to study the relevant literature from 2012 to 2016, summarizing more key themes [12].

The inventory model for perishable goods usually assumes a constant deterioration rate [13,14]. The inventory model for perishable products also usually assumes a constant spoilage rate. However, due to the significant impact of temperature and microorganisms on product quality, the spoilage of perishable products becomes uneven over time [15]. Researchers analyzed the decay rate related to time; it can be divided into decay rates that are linearly proportional to time [16,17] and decay rates as a Weibull function with three parameters [18].

Shiue [19] established an EOQ model that allows for out-of-stock situations under the assumption of deterministic demand, instant delivery with quantity discounts, and a certain probability distribution of the spoilage time of perishable goods. Levin et al. [20] observed that a large number of consumer goods being displayed in supermarkets leads to customers purchasing more goods. The more inventory is displayed, the more it stimulates consumption, and this phenomenon is summarized as demand relying on inventory. Urban [21] summarized two inventory control models, with one being that the demand rate of an item depends on the initial inventory level and the other being that the demand rate of an item is a function of the instantaneous inventory level. Maity and Maiti [22] established an inventory model for items, taking into account similar and different supplementary items. The classic EOQ model solves the problem of inventory replenishment for a single product. As the replenishment of multiple items is a common situation in operational environments, the replenishment of multiple products has become a research hotspot for scholars [23–37].

Automated pricing and replenishment strategies are both algorithmic and data-driven marketing strategies aiming to maximize revenues and profits, which automatically adjust replenishment strategies based on inventory levels and sales trends, among many other factors. The strategies automatically calculate the optimal replenishment volume and time by analyzing historical sales data and forecasting future demands to ensure that products are in sufficient supply while avoiding inventory overstock and waste. The automatic

pricing and replenishment strategies of goods play a very important role in the operation of supermarkets. In supermarkets selling raw and fresh products, the shelf life of vegetables in general is relatively short, the quality deteriorates with the passing of sales time, and most varieties cannot be sold the next day if they are not sold in one day. Therefore, supermarkets usually replenish stocks on a daily basis based on the historical data of sales and demand of each product.

Due to the fact that a large variety of vegetables sold in supermarkets have different origins, and that the wholesale time is usually between 3:00 and 4:00 in the morning, merchants must make replenishment decisions for each vegetable category on the same day without knowing exactly the sale and wholesale price of a specific product. The pricing of vegetables generally adopts the "cost-plus pricing" method, and supermarkets usually sell goods with poor appearances or damages caused in transportation at a discount. Reliable analysis of market demand is especially important for replenishment and pricing decisions. From the demand side, there is often a certain correlation between the sales volume and time of vegetable products. From the perspective of the supply side, the supply of vegetables is more abundant from April to October, and the limitation in supermarket sales space makes a reasonable product portfolio extremely important.

Considering that supermarkets make replenishment plans based on categories, this paper needs to analyze the relationship between the total sales volume of each vegetable category and cost-plus pricing, and then to give the daily capture volume and pricing strategy of each category in the coming week (1–7 July 2023) so as to maximize the revenue of supermarkets.

In this paper, we implement the automatic pricing and replenishment strategy of perishable products with a time-varying deterioration rate based on an improved SVR-LSTM ARIMA hybrid model. If the sales price and cost can be calculated with categories as units, the ratio of cost-plus pricing can be obtained. Then, we can match the total sales volume with the cost-plus pricing ratio through the least-squares method. The practical calculation shows that the conditions for a strong linear correlation are not satisfied, so the non-parametric support vector regression (SVR) method is used for fitting according to the final results. For the prediction after 7 days considering the data in the last 30 days, the periodic time information is extracted through the long-term and short-term recurrent neural network and prior knowledge for weighting, on the premise that the supply and demand relationship of the supermarket products is ensured. The sales volume of the supermarket is then calculated. The cost-plus ratio is calculated via the above-mentioned trained SVR model so as to maximize the benefit of the supermarket.

Considering the limited space for the sales of vegetable products, supermarkets hope to further formulate a replenishment plan for single products, control the total number of saleable products at 27–33, and output the replenishment volume and pricing strategy of single products on 1 July so as to maximize the revenue of supermarkets on the premise of meeting the maximum market demand for various categories of vegetables.

This article only considers the varieties that can be sold between 24 and 30 June. The conditions that need to be met are as follows:

(a)  Ensure that the market demand for various categories of vegetable commodities is met;
(b)  A correction of the shelf life is made in consideration of the damage;
(c)  The average order quantity of each single product is greater than 2.5 kg;
(d)  The total gain is maximized.

This paper carries out multi-objective planning and optimal estimation of these four tasks. Based on the strategy of the greedy algorithm, as many items with higher unit profits are placed in the "backpack" as possible. If the backpack is not full or too full to be filled in, it is corrected by backtracking.

In order to solve the above problems, the research on the automatic pricing and replenishment strategy selected perishable products of a specific category of vegetables, and the sales data of cyclical and non-cyclical vegetables in offline supermarkets were

analyzed through the automatic pricing of vegetables. This was then extended to the research of perishable products with a time-varying spoilage rate. Based on the above analysis, the automatic pricing and replenishment strategy of perishable goods with time-varying deterioration rates can be modeled and designed. The data for this paper come from the 2023 Higher Education Society Cup National College Student Mathematical Modeling Competition Questions of China at http://www.mcm.edu.cn/html_cn/node/c7 4d72127066f510a5723a94b5323a26.html (accessed on 7 September 2023).

This paper is organized as follows: Section 2 presents related models. Section 3 reports the results obtained. Finally, conclusions are presented in Section 4.

## 2. Related Models

This paper focuses on the modeling of automatic pricing and replenishment strategies for perishable products with time-varying deterioration rates based on an improved SVR-LSTM-ARIMA hybrid model.

### 2.1. Model 1: SVR Model

Support vector regression (SVR) aims to find a line that can make all points as close to it as possible in order to make predictions about the data. Because the relevant data in this paper do not satisfy a strong linear correlation, we need to use non-parametric support vector regression (SVR) methods to fit the relevant data.

The SVR model creates a "spaced band" on both sides of the linear function; the space between "spaced bands" is $\varepsilon$ (this value is often given empirically); the loss is not calculated for all the samples that fall into the spaced band, that is, only the support vector affects its function model; and finally, the optimized model is obtained by minimizing the total loss and maximizing the space interval [38]. $f(x) = wx + b$ is the final model function required in this paper. $f(x) + \varepsilon$ and $f(x) - \varepsilon$ are the upper and lower edges of the spaced band; $\xi^*$ is the difference between the projection of the sample point below the lower edge of the spaced band and the $y$ value of the sample point. The formula is expressed as follows in Equations (1) and (2):

$$\begin{cases} \xi_i = y_i - (f(x_i) + \varepsilon), \ y_i > f(x_i) + \varepsilon \\ \xi_i = 0, otherwise \end{cases} \tag{1}$$

$$\begin{cases} \xi_i^* = (f(x_i) + \varepsilon) - y_i, y_i < f(x_i) + \varepsilon \\ \xi_i^* = 0, otherwise \end{cases} \tag{2}$$

### 2.2. Model 2: Long Short-Term Memory (LSTM) Model

Long short-term memory (LSTM) is a type of temporal recurrent neural network. LSTM is specifically designed to address the long-term dependency problem of general RNNs (recurrent neural networks), and all RNNs have a chain form of repetitive neural network modules. We use the sales data from the past 30 days to fit using a long short memory recurrent neural network (LSTM), and use the average MSE as the loss function of the model to predict the 7-day data to obtain the sales volume.

The first step of the LSTM is to determine what information can pass through the cell state. This decision is controlled by the "forget gate" layer via the sigmoid function, which generates a value of 0 to 1 based on the output of the previous moment $h(t-1)$ and the current input $x(t)$, deciding whether to let the information learned at the previous moment $C(t-1)$ pass or partially let it pass [39].

$$f_t = \sigma(W_f[h_{t-1}, x_t] + b_f) \tag{3}$$

The second step is to generate new information that needs to be updated in this article.

$$i_t = \sigma(W_i[h_{t-1}, x_t] + b_i) \tag{4}$$

$$\widetilde{C}_t = \tanh(W_C[h_{t-1}, x_t] + b_C) \tag{5}$$

A memory gate is a control unit used to control whether or not data at time $t$ (current) are incorporated into the cell state. First, the tanh function layer is used to extract the valid information in the current vector, and then the sigmoid function is used to control how much of these memories should be put into the unit state.

$$C_t = f_t C_{t-1} + i_t \widetilde{C}_t \tag{6}$$

The final step is to determine the output of the model; then, by multiplying it by pairs with the output from the sigmoid function, we obtain the output of the model.

$$o_t = \sigma(W_O[h_{t-1}, x_t] + b_O) \tag{7}$$

$$h_t = o_t \tanh(C_t) \tag{8}$$

*2.3. Model 3: Time Series (ARIMA)*

The ARIMA model attempts to extract time series patterns hidden behind data through autocorrelation and differentiation, and then it uses these patterns to predict future data. We use the ARIMA model to predict prices for the next day using past data.

ARIMA models include autoregressive (AR) models and moving-average (MA) models. The AR model describes the relationship between current and lagging values and uses historical data to predict future values. The MA model makes use of a linear combination of past residual terms to observe future residuals [40]. The ARIMA prediction model can be written as the following formula:

$$\hat{p}^{\{t\}} = p_0 + \sum_{j=1}^{p} \gamma_j p^{\{t-j\}} + \sum_{j=1}^{q} \theta_j \varepsilon^{\{t-j\}} \tag{9}$$

Here, $p$ is the order of the autoregressive model (AR) and $q$ is the order of the moving-average model (AM). $\varepsilon$ is white noise, and $\varepsilon$ has a mean of 0 and a variance of 1. $\gamma_j$ and $\theta_j$ are the fitting coefficients. $p_0$ is the constant term. Respectively, ACF (autocorrelation function) and PACF (partial correlation function) are both functions that evaluate the linear relationship between historical data and current values.

$$ACF(q) = \frac{Cov(X_j, X_{j-q})}{Var(X_0)} = \frac{\frac{1}{n-q}\sum_{j=q+1}^{n}(x_{j-q} - \overline{x})}{\frac{1}{n}\sum_{j=1}^{n}(x_j - \overline{x})^2} \tag{10}$$

## 3. Results

First, we process and analyze the relevant data. Then, we use an improved hybrid model for correlation analysis and draw some meaningful conclusions. Finally, we analyze the advantages of our model in this section. We select perishable vegetables as the research category and calculate the cost-plus ratio for each vegetable category. A correlation analysis is conducted with the total sales, and a non-parametric relationship curve is obtained using support vector regression for nonlinear fitting. The long and short memory recurrent neural network is then used to predict sales volume, and a pricing strategy is calculated based on the fitting curve. Additionally, this paper establishes a correlation between loss rate and shelf life, corrects the daily average sales volume index, and solves the problem of quantity and category of replenishment using a backpack problem approach. By considering multiple constraints, a quantitative category replenishment volume and pricing strategy is obtained.

*3.1. Data Processing*

After merging the collected commodity information and detailed sales data of six vegetable categories, the data are cleaned and quantified, and then the characteristics are selected and transformed to establish the model. In this section, a box plot is used for data cleaning; its biggest advantage is that it is not affected by outliers, so it can accurately and stably depict the discrete distribution of data, quickly determine whether the data contain outliers, and understand the degree of skew and distribution range of the data. We use the filtered data to draw a box plot and save it to the output directory. In this way, outlier handling can be automated to reduce manual intervention in data processing, identify more objectively outliers in the data, and accurately calculate the number of outliers. Only six vegetable categories are shown in the box plot of the raw data, as shown in Figure 1.

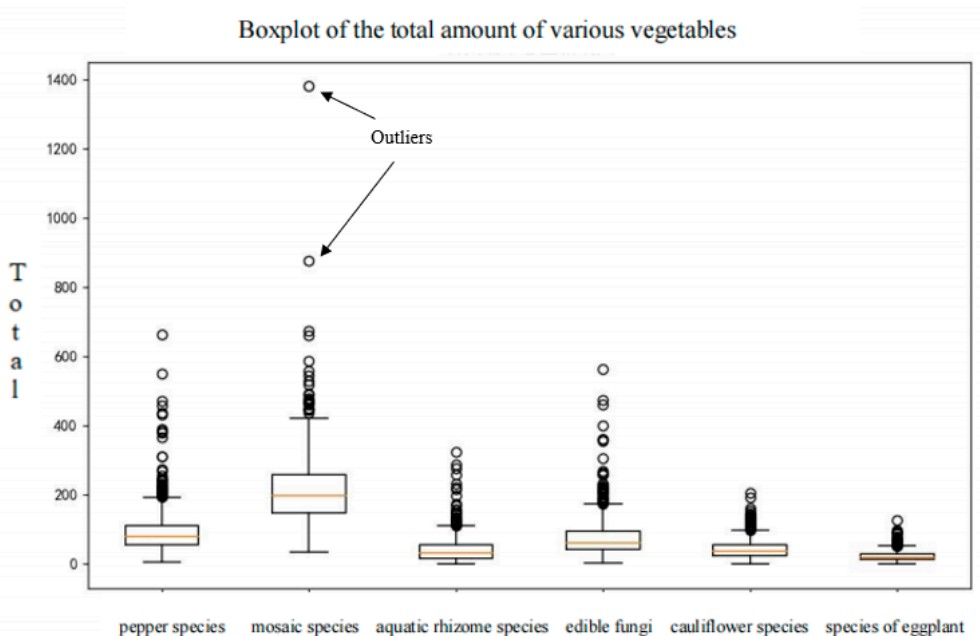

**Figure 1.** Box plot of sales volume.

According to the analysis of Figure 1, only some of the data have outliers. After eliminating the outliers, there is a difference in the number of missing values between different features. In this case, after careful analysis of the dataset, KNNImputer is used to fill in the missing values.

First, the calculation of the Euclidean distance with the missing values is as follows:

$$d_{ij} = W * D \tag{11}$$

$$W = T/P \tag{12}$$

The value $d_{ij}$ represents the Euclidean distance. $W$ represents the weight. $D$ represents the square distance from the current coordinates. $T$ represents the total number of axes, and $P$ represents the number of current axes.

Then, the nearest-neighbor samples are found through the Euclidean distance matrix and the mean of the non-null values of the corresponding positions of the nearest-neighbor samples is used to fill in the missing values.

In the program designed for this article, the n_neighbors parameter is set to 5, that is, for each missing value, the article fills it with the last five non-null values.

The filling results are shown as follows in Figures 2 and 3. (using cauliflower and edible mushrooms as examples):

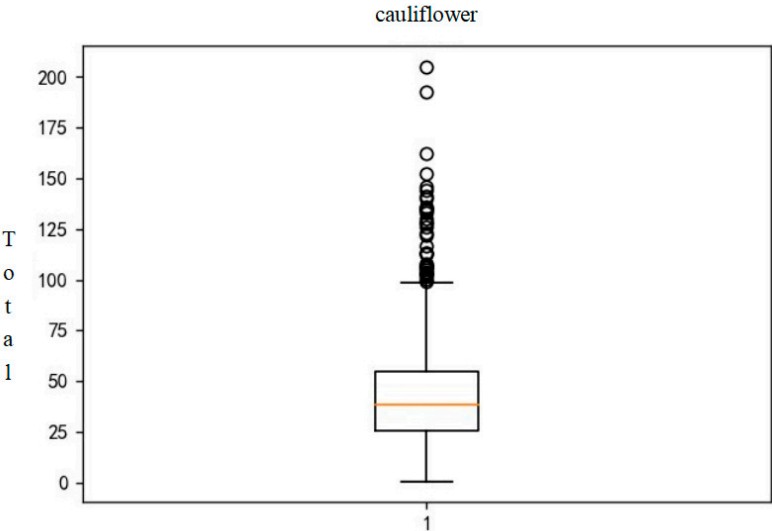

**Figure 2.** Box plot of cauliflower sales.

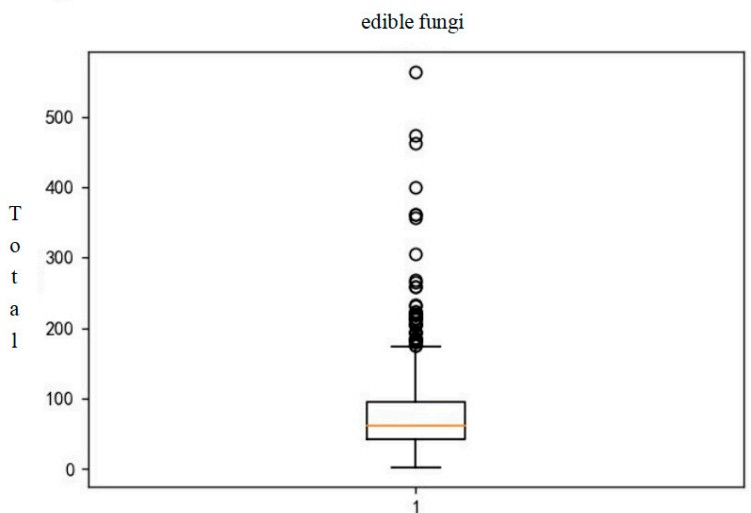

**Figure 3.** Box plot of edible mushroom sales.

Here, we take the first seven products of chili peppers and edible mushrooms as examples, as shown in Figures 4 and 5.

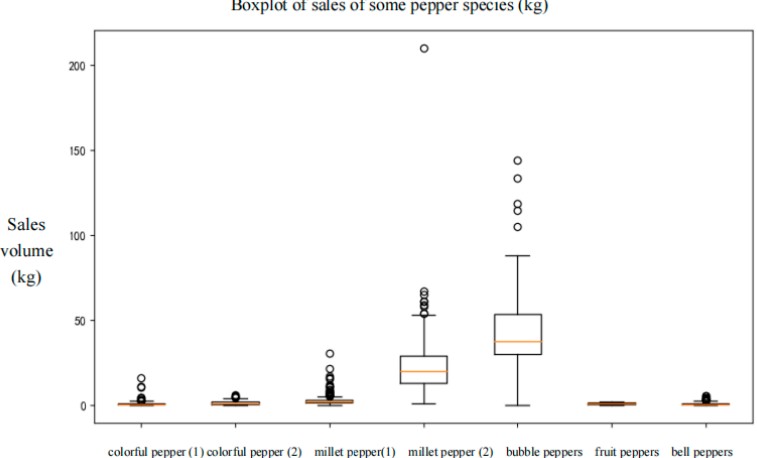

**Figure 4.** Box plot of the sales volume of each chili pepper product.

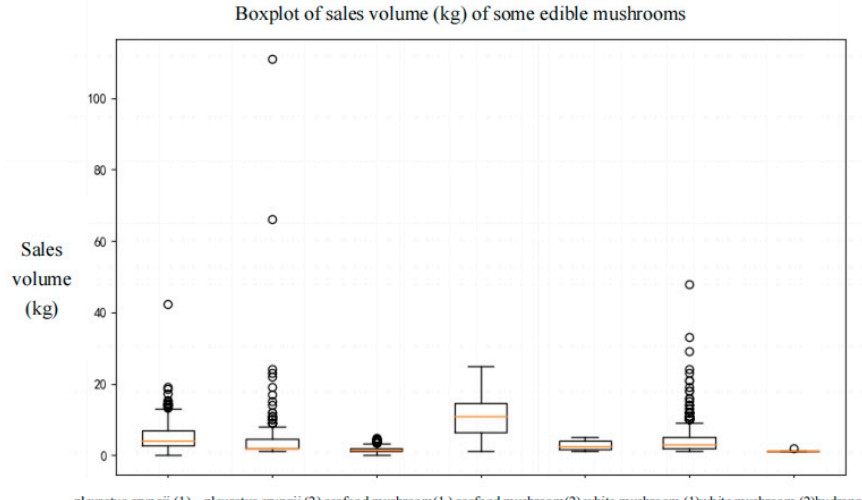

**Figure 5.** Box plot of the sales volume of each edible mushroom product.

The above is the preprocessing process of the relevant data, but when this article processes data in a large category, there are still outliers in the small category. This may be due to a variety of factors, such as data heterogeneity, sample sizes and reliability, specific characteristics, and context, as well as the choice in processing methods, which is not further explored in this article. Of course, in order to better deal with outliers in small categories, it is recommended to further study and understand the characteristics of small categories and to adopt appropriate outlier-handling methods to improve the quality and accuracy of the data.

Through data description and analysis, we first come to the basic understanding of each category of vegetable. The data are visualized. The data patterns and trends are intuitively observed and the distribution trend of the sales volume of various categories of vegetables is carefully analyzed, as shown in Figures 6–8.

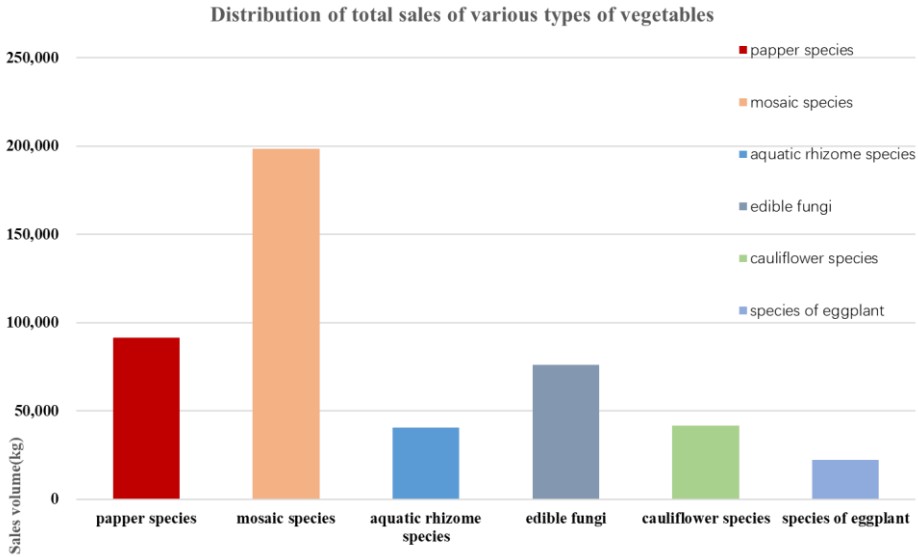

**Figure 6.** Distribution of total sales of each category of vegetables.

**Proportion of total sales of various types of vegetables**

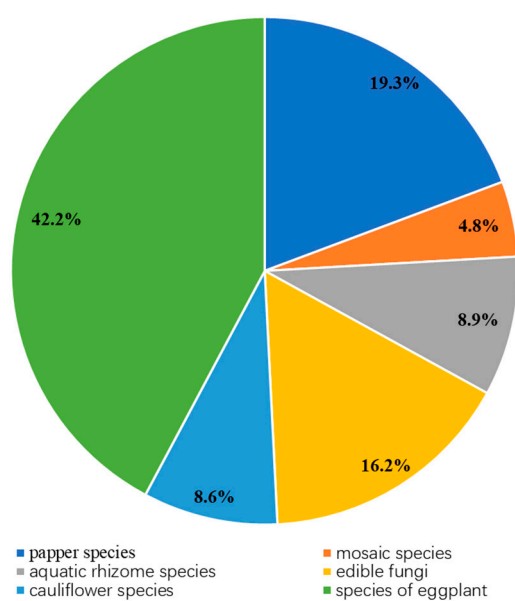

**Figure 7.** Proportion of total sales of vegetables by category.

**Top 10 best-selling vegetable items**

**Figure 8.** Distribution of the top 10 vegetable items with the highest sales volume.

From the above chart, it can be seen that in terms of category sales, vegetables with "flowers and leaves" witness the highest sales volume, followed by "peppers", with "eggplant" having the lowest. In terms of single-product sales, "Wuhu Green Pepper (1)" has the highest sales volume, followed by "Broccoli" and "Net Lotus Root (1)".

Next, this article further analyzes the change in sales volume over time, and whether there is a certain trend. First, this paper uses statistical tools to obtain the trend of morning and afternoon sales of each category of vegetables, and then statistically analyzes the changes in the number of vegetables in each quarter (at an interval of 3 months), as shown in Figures 9 and 10.

**Distribution of morning and afternoon sales volume of vegetables by category**

**Figure 9.** Distribution of morning and afternoon sales of different vegetable categories.

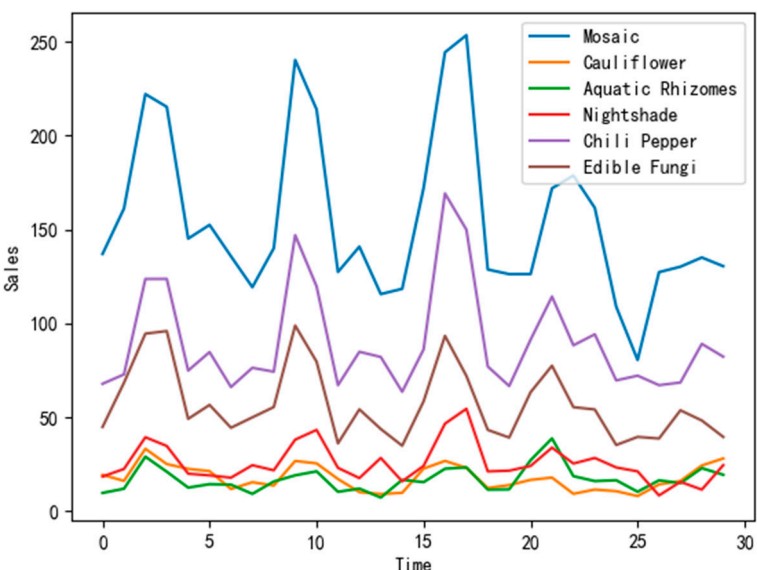

**Figure 10.** Line chart of the number of vegetables in different time periods.

In this paper, we can observe that the sales volume is high in the afternoon, and the sales volume of each category has peaks and bottoms throughout the year. There are obvious seasonal changes, which displays the time trends in the sales volume. In addition, the sales volume of the categories of vegetables with flowers and leaves fluctuates the most, which may be affected by various factors such as seasonality, climatic factors, holidays, and celebrations, as well as market demand and consumption habits.

First, we consider the cost-plus pricing and calculate the total price, total profit, and cost-plus ratio, as shown in Figure 11.

$$S_{\text{Total sales volume}} = \sum n P_{\text{sell}} \tag{13}$$

$$S_{\text{profit}} = \sum n (P_{\text{sell}} - P_{\text{whole sale}}) \tag{14}$$

$$\alpha = \frac{S_{\text{Total sales volume}}}{S_{\text{Total sales volume}} - S_{\text{profit}}} - 1 \tag{15}$$

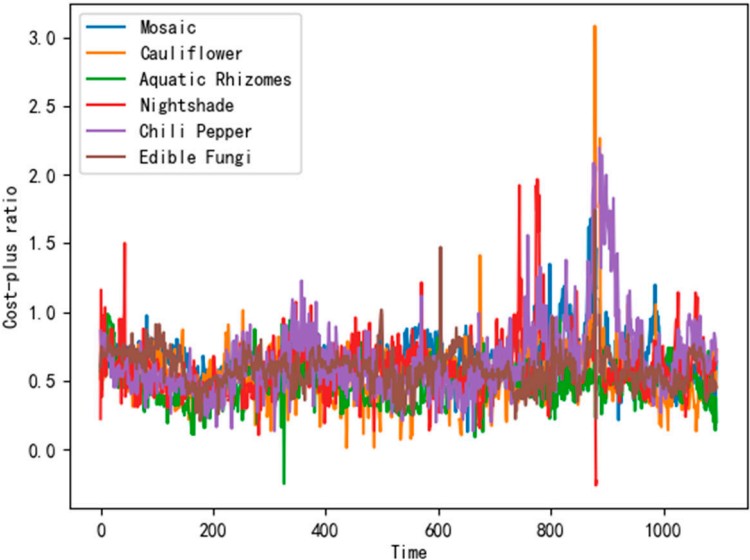

**Figure 11.** Variation of cost-plus ratio by category over time.

Figure 11 shows that the daily sales of the six vegetables have peaks and bottoms from July 2020 to 30 June 2023, showing obvious seasonal fluctuations.

### 3.2. Model 1—Solving Process

Considering that the sales volume is a typical variable that fluctuates with time, this paper selects the day as the unit, accumulates the sales data of all categories in a day, and plots the curve of the sales volume of each category over time. Since the dataset collected during the time contains data from over three years, the black dotted line is used to divide the years. It is clear from Figure 12 that the sales volume of all types of vegetables shows obvious seasonality, especially for edible fungi and aquatic rhizomes. In this article, the SVR model is selected, as shown in Figure 12.

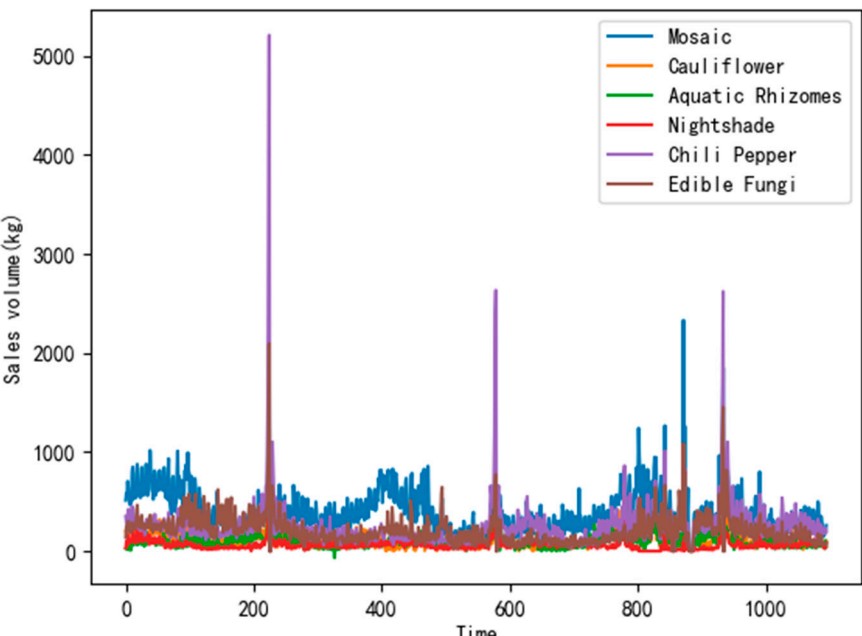

**Figure 12.** Variation in sales volume of each category over time.

Considering that the sales volume is a typical variable that fluctuates over time, we select the day as the unit, accumulate the sales data of all categories in a day, and plot the change curve of the first sales volume of each product over time. Since the datasets collected during the time include data from over three years, the black dotted line is used to divide the years. It can be clearly seen that the sales volume of all types of vegetables shows obvious seasonality, especially edible fungi and aquatic rhizomes, as shown in Figure 13.

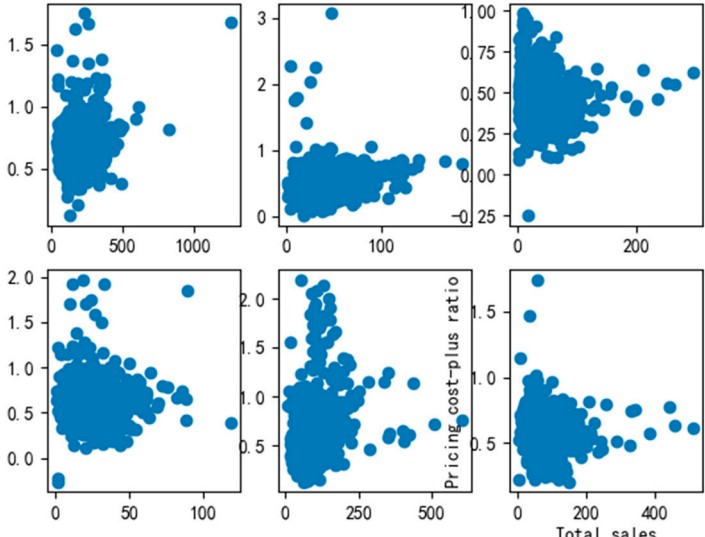

**Figure 13.** The relationship between the total sales volume of each category and the cost-plus pricing.

The correlation coefficient is as shown in Table 1.

**Table 1.** Correlation coefficients.

| Category | Correlation Coefficient |
| --- | --- |
| Aquatic rhizomes | −0.198 |
| Flowers and leaves | 0.185 |
| Broccoli | 0.136 |
| Eggplant | 0.057 |
| Peppers | 0.272 |
| Edible fungi | −0.090 |

The SVR model creates a "spaced band" on both sides of the linear function; the space between the "spaced bands" is $\varepsilon$ (this value is often given empirically); the loss is not calculated for all the samples that fall into the spaced band, that is, only the support vector affects its function model; and finally, the optimized model is obtained by minimizing the total loss and maximizing the space interval. $f(x) = wx + b$ is the final model function required in this paper based on Equations (1) and (2).

There are also some differences between the SVR model and the traditional general linear regression model, and the differences are mainly reflected in the following.

(a)   In the SVR model, the loss is calculated if and only if the absolute value of the difference between and $f(x)$ and $y$ is greater than $\varepsilon$, whereas in the general linear model, the loss is calculated as long as $f(x)$ and $y$ are not equal.

(b)   The optimization methods of the two models are different. In the SVR model, the model is optimized by maximizing the width of the spaced band and minimizing the loss, while in the general linear regression model, the model is optimized by the average value after gradient descent.

Finally, the SVR regression fitting algorithm is carried out on the data of each category, with the non-parametric relationship curve obtained as shown in Figure 14.

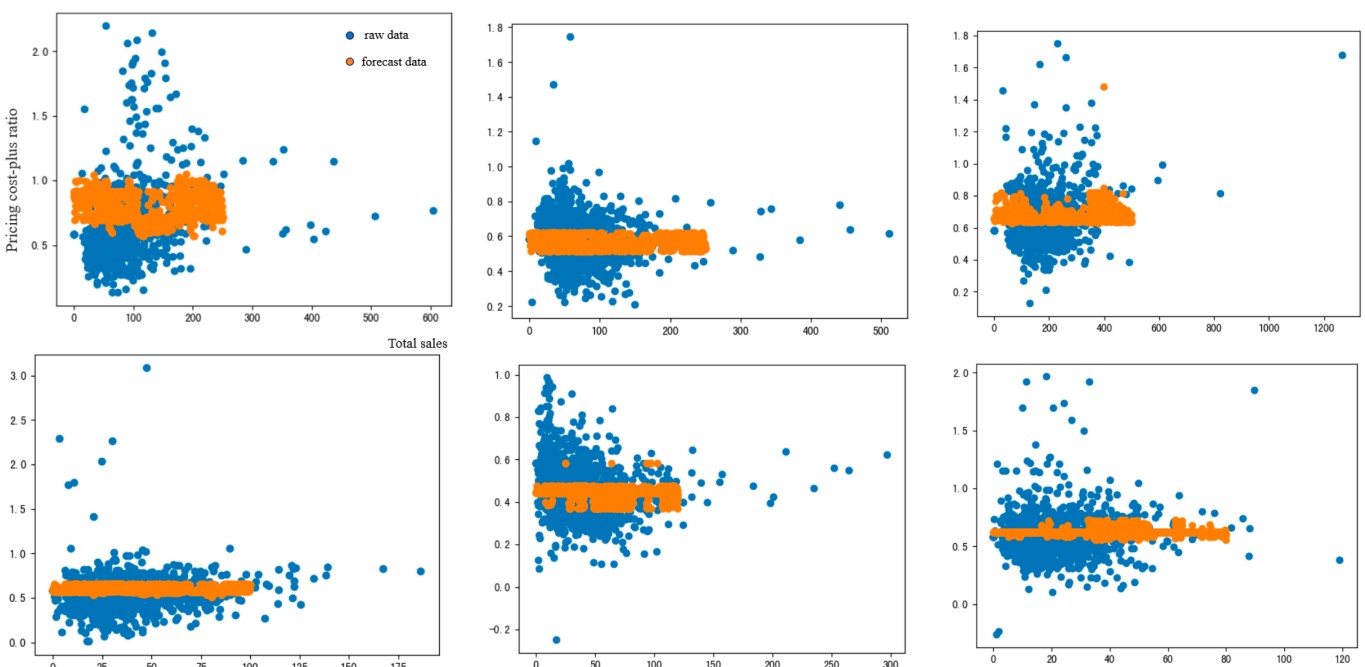

**Figure 14.** SVR non-parametric regression fitting of each category's data. (Blue scatters: These scatters represent the raw data of the first set of data; Orange scatters: These scatters represent data predicted using the support vector regression model).

The fitting results basically fluctuate around the mean, which is greatly affected by time and other factors.

*3.3. Model 2—Solving Process*

Next, this article examines the sales volume and cost-plus ratio over the studied time period. In time series forecasting, LSTM can be used as both a multivariate prediction mechanism and a unit prediction mechanism, and it has the ability to memorize long- and short-term memory. Thus, the LSTM (long short-term memory) model is used for analysis. Figure 15 is derived from the data analysis.

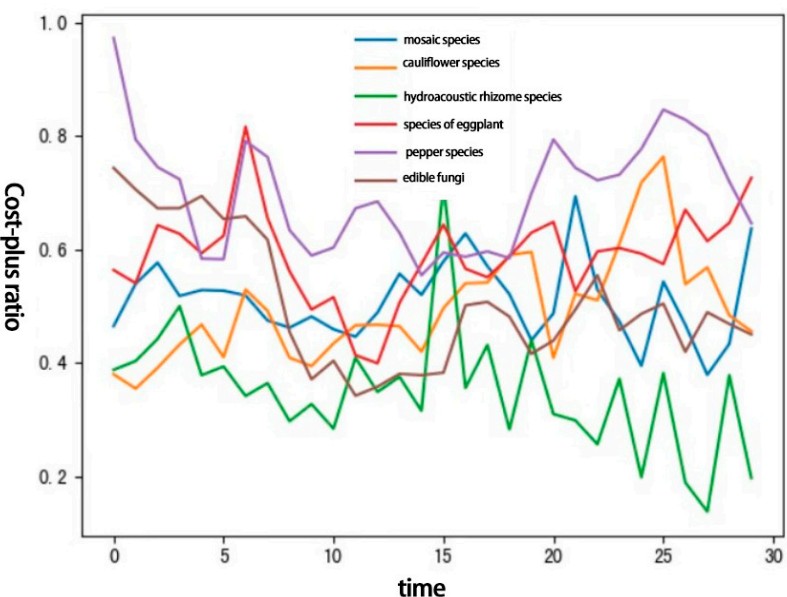

**Figure 15.** Cost-plus ratio over the last 30 days.

In the last 30 days, the consumption of various categories was relatively stable, and considering that 1–7 July 2023 was a complete week and there was no statutory holiday, working days and weekends also significantly affected the sales volume. On the one hand, we ensure that the supermarket has the greatest revenue. On the other hand, meeting sales demand is the first priority. If one blindly pursues a higher rate of return, which results in insufficient or excessive sales space, greater side effects occur, such as the expiration of the products, the insufficient inventory capacity of the warehouses, and the inability to meet the consumption needs of consumers. These effects spoil consumers' loyalty to the supermarket and can even cause them to permanently lose customers.

In this program, the sales data in the last 30 days are fitted through the long and short-term memory recurrent neural network (LSTM), and the average MSE is used as the loss function of the model to directly predict the data in the next 7 days to obtain the sales volume M1, M2,..., M7., In order to obtain the cycle information, the average sales volume in 30 days is directly used to calculate the period-weighted β1, β2,..., β7, and the weighted number is obtained through the data of the same period in previous years; this is shown in Table 2.

**Table 2.** Weights.

| Date | Numeric Value |
|---|---|
| 1 July | 0.175 |
| 2 July | 0.127 |
| 3 July | 0.110 |
| 4 July | 0.130 |
| 5 July | 0.143 |
| 6 July | 0.159 |
| 7 July | 0.155 |

The weighting of the number of days of the week leads to A1, A2,... A7, with the specific calculation method as follows:

$$A_n = Mean \text{ (Sales within 30 days)} \beta_n, n = 1, 2, \cdots, 7 \tag{16}$$

The final prediction results are linearly weighted:

$$D_n = 0.5M_1 + 0.5A_n, n = 1, 2, \cdots, 7 \tag{17}$$

Finally, the forecasted sales volume and pricing strategy are derived.

*3.4. Model 3—Solving Process*

Since prices are seasonal, time series models can also be used for forecasting. Time series (ARIMA), or autoregressive integer moving average, is a statistical analysis model that uses time series data to predict future trends. The basic idea of ARIMA is that the data sequence formed by predictions is treated as a random sequence over time, and a model can be used to approximate the description of that sequence. Once this series is determined, the model can predict future values based on the past and present values of the time series. With this model, this paper attempts to predict the future unit price of a commodity based only on the unit price data as of the current month.

For the price q time series, with the help of the ARIMA model, this paper uses past data to make a simple prediction of the next day's prices. This article optimizes the grid strategy so that it can move the grid based on the predictions given by ARIMA. In the model in this paper, the movement of the grid is determined by the weighting and summing of the long-term indicators in the MA and the short-term indicators in the ARIMA model [39]. The amount of movement of the grid is calculated using the following formula:

$$\overline{p}_i^{\{t\}} = p_i + \omega(MA^{\{t\}}(N) - MA^{\{t-1\}}(N)) + \mu(ARIMA^{\{t+1\}} - ARIMA^{\{t\}}) \tag{18}$$

Parameters $\omega$ and $\mu$ control the weighting of the two metrics during grid movement. These two parameters are be adjusted adaptively during the backtesting phase, and this article initializes $\omega$ to 0.3 in the first cycle. Therefore, the final decision is made by an adjusted grid model, which predicts the price of the commodity. This is shown in Figure 16.

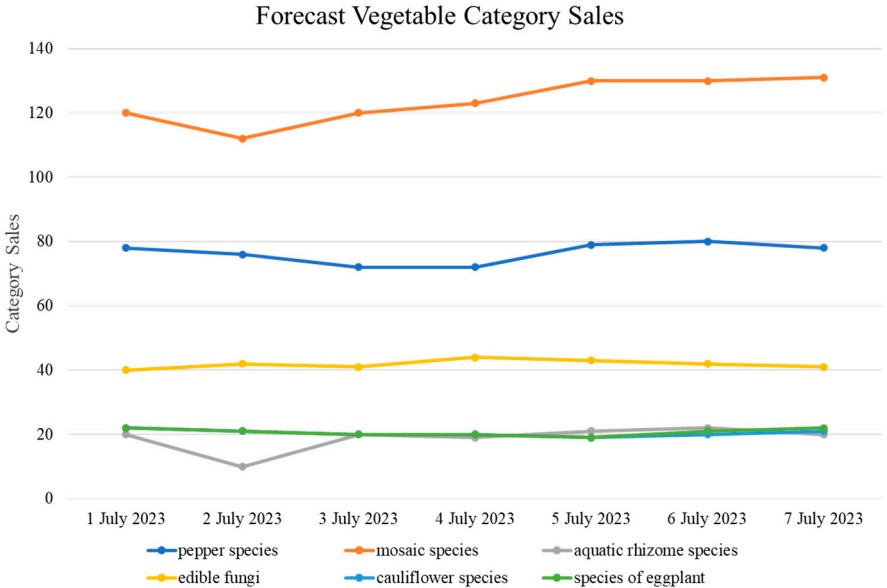

**Figure 16.** Forecasted sales volume of vegetable categories.

The fitting results basically fluctuate around the mean, which is greatly affected by time and other factors. In order to predict the total daily replenishment and pricing strategy of each vegetable category in the coming week, we first look at the percentage of sales volume in that time period.

$$C = W \times (1 + L/100) \tag{19}$$

In the above equation, $C$ denotes the cost per kilogram of the vegetable variety; $W$ represents the wholesale price of the vegetable categories; and $L$ denotes the recent loss rate of the vegetable products. We determine the markup rate based on the expected sales volume as shown in Table 3:

**Table 3.** The markup rate based on the expected sales volume.

| The Expected Sales Volume | The Markup Rate |
|---|---|
| <20 kg | 150% |
| <50 kg | 130% |
| Else | 120% |

Products of low sales volume aim to obtain a higher yield per unit of goods, while products of higher sales volume aim to increase their sales through lower prices.

Step 3: We calculate the sales price for each category.

$$S = C \times M \tag{20}$$

In the above formula, $S$ denotes the selling price per kilogram of the vegetable products; $C$ denotes the cost per kilogram of the vegetable products; and $M$ denotes the markup rate. The cost (per kilogram) of each vegetable category is calculated in Table 4.

**Table 4.** Cost of each vegetable category.

| Category | Cost |
|---|---|
| Aquatic rhizomes | 11.22 yuan |
| Flowers and leaves | 5.44 yuan |
| Broccoli | 7.57 yuan |
| Eggplant | 5.17 yuan |
| Peppers | 7.61 yuan |
| Edible fungi | 7.20 yuan |

*3.5. Final Results*

In this paper, a histogram is used to visualize the following data:

1.  Understanding the distribution: The histogram can clearly show the distribution of the average revenue of each item. By looking at the shape of the histogram, it is possible to understand the concentrated trend in the average return, the extent of the dispersion, and whether there are outliers.
2.  Identifying the median and mean: The central trend in the histogram can help identify the mean and median. This can be very helpful in understanding overall trends and evaluating the performance of the product portfolio.
3.  Outliers found: The tail of the histogram (long tail or short tail) may indicate the presence of outliers. Outliers can be extreme positive or negative gains, and with histograms; these extremes can be more easily identified.
4.  Comparing different items: In case of a histogram of multiple items, one can conveniently compare the average revenue distribution between the items. This can help find the best- or worst-performing pieces, as well as understand the differences between different pieces.

In addition, in order to develop a pricing strategy, it is necessary to consider the wholesale price of each category, the average loss rate, and the expected sales volume to calculate the cost of each vegetable category (per kilogram). This is shown in Tables 5 and 6.

**Table 5.** Forecast sales volume.

| Date | Flowers and Leaves | Broccoli | Aquatic Rhizome Species | Eggplant | Peppers | Edible Fungi |
|---|---|---|---|---|---|---|
| 1 July | 152.9374 | 19.79373 | 20.42285 | 23.27763 | 95.03952 | 54.13958 |
| 2 July | 110.9889 | 14.36459 | 14.82115 | 16.89291 | 68.97154 | 39.28986 |
| 3 July | 96.13208 | 12.44177 | 12.83722 | 14.63165 | 59.73913 | 34.03059 |
| 4 July | 113.6106 | 14.70391 | 15.17126 | 17.29195 | 70.60079 | 40.21979 |
| 5 July | 124.9717 | 16.1743 | 16.68839 | 19.02115 | 77.66087 | 44.23977 |
| 6 July | 138.9546 | 17.98401 | 18.55562 | 21.14939 | 86.3502 | 49.18967 |
| 7 July | 135.4588 | 17.53159 | 18.08881 | 20.61733 | 84.17786 | 47.9522 |

**Table 6.** Pricing policies.

| Date | Flowers and Leaves | Broccoli | Aquatic Rhizome Species | Eggplant | Peppers | Edible Fungi |
|---|---|---|---|---|---|---|
| 1 July | 0.641624 | 0.48805 | 0.491309 | 0.573781 | 0.64409 | 0.584361 |
| 2 July | 0.64675 | 0.512502 | 0.524329 | 0.573377 | 0.60185 | 0.598898 |
| 3 July | 0.629628 | 0.522525 | 0.537349 | 0.576475 | 0.596621 | 0.598996 |
| 4 July | 0.645114 | 0.510706 | 0.522049 | 0.572993 | 0.602605 | 0.598604 |
| 5 July | 0.64025 | 0.503097 | 0.512382 | 0.571804 | 0.607352 | 0.596339 |
| 6 July | 0.638909 | 0.494664 | 0.501243 | 0.571689 | 0.620462 | 0.591398 |
| 7 July | 0.638808 | 0.496633 | 0.503928 | 0.57154 | 0.616203 | 0.592846 |

The following is a data analysis for a single product to calculate the average revenue for 251 single item types (Figure 17).

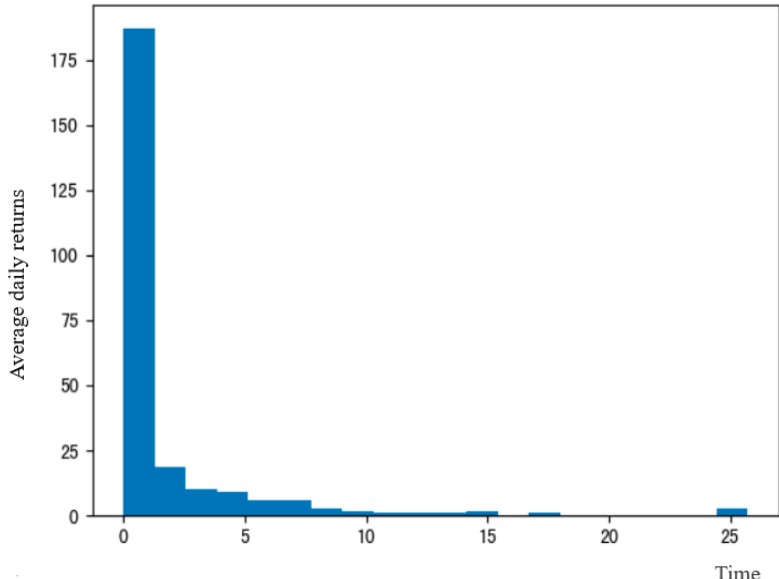

**Figure 17.** Histogram of average revenue of each single product.

The vast majority of single products generate an average daily profit of less than CNY 10, but there are also single products with higher returns. The average unit price and the amount of returns are further analyzed, as shown in Figure 18.

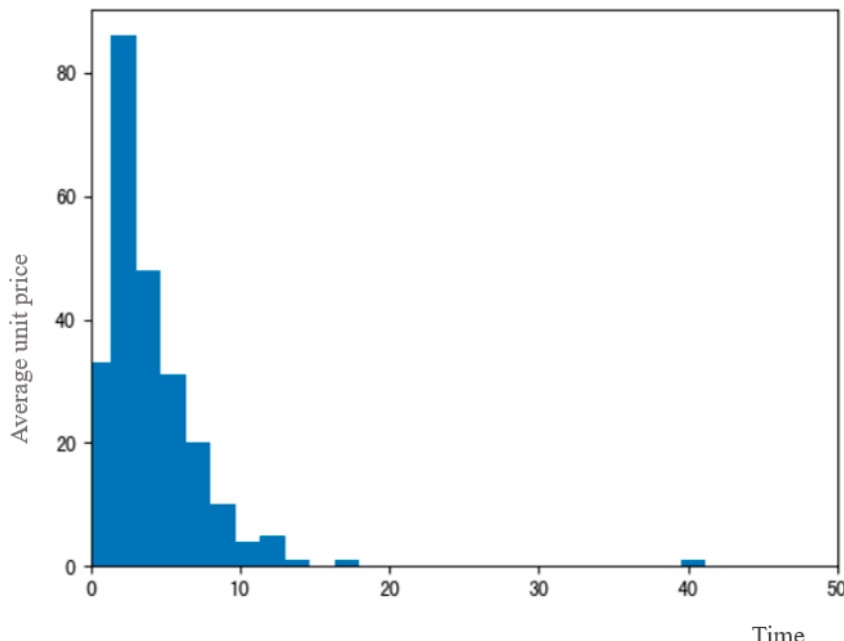

**Figure 18.** Histogram of the average unit price of each single product.

By filtering the data, it is possible to filter out four items that have no transaction information, which are represented by local vegetables.

In order to meet the actual order needs, we require the order quantity of each single product to meet the requirements of a minimum display weight greater than 2.5 kg. The loss rate data are introduced to correct the shelf life, and the correlation between the loss

rate and the shelf life is established to further filter the single products that do not meet the requirements.

Firstly, the average value of the loss rate is 9.43%, and the standard deviation is 0.052, as is shown in Figure 19.

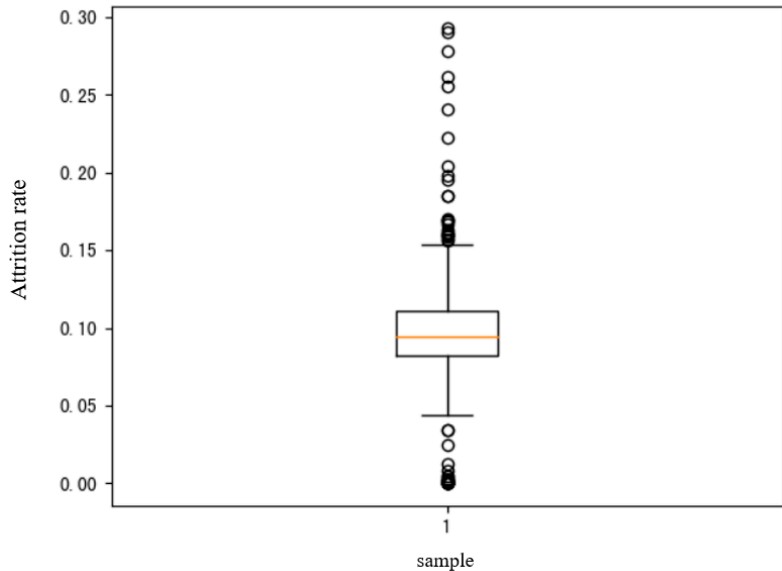

**Figure 19.** Box plot of loss rate.

The formula is derived from the loss data:

$$B = \begin{cases} 1, S > 9.43\% \\ -\frac{S}{S_m} + 2, S \leq 9.43\% \end{cases} \tag{21}$$

The loss rate is $S$. The average loss rate of the sample is $S_m$. $B$ is the corrected shelf life, and the loss rate is $\geq 0$. The modified shelf life is directly applied to the average daily sales volume, i.e., only $B \times M > T$ needs to be met. The histogram of corrected average number is shown in Figure 20.

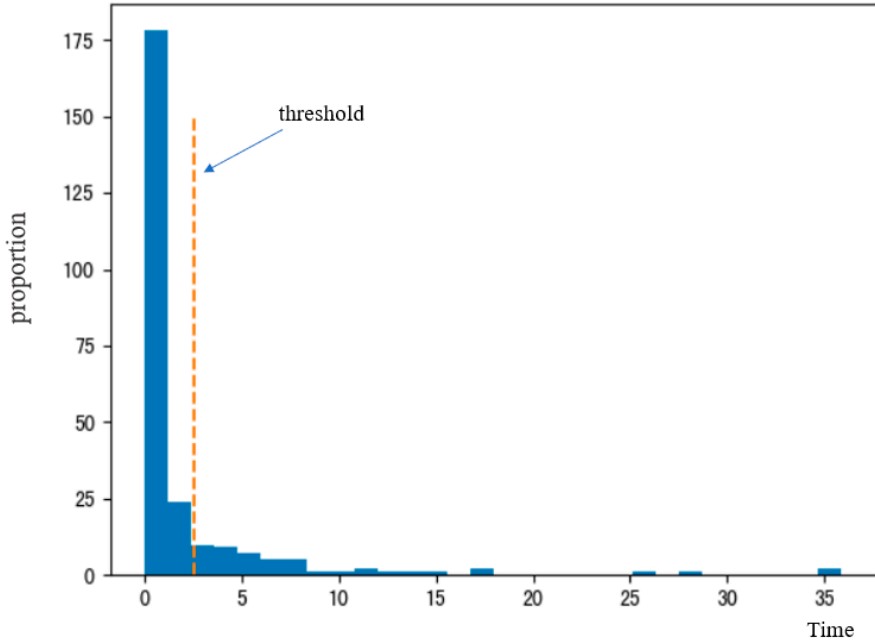

**Figure 20.** Histogram of the corrected average quantity.

Based on this, 198 items were screened out, and only 49 categories of items were left to choose from, among which the dotted line drawn represents the threshold position.

In the end, one only needs to consider the saleable items on 24–30 June and continue to screen the items to be selected. Saleable products on 24–30 June amounted to a total of 61 items, and after a comprehensive comparison, we continued to screen out 13 items; finally, only 36 items were left to choose from.

Considering that the total number of categories is backpacks and that the single items in each category are calculated independently, six backpacks are adequate if they can be filled. In this case, it is proven that the greedy algorithm can achieve the globally optimal solution while having better performance.

Of course, in order to make better replenishment and pricing decisions for vegetable commodities, on top of providing daily fluctuating cost prices, purchase quantities, purchase prices, and loss rates, supermarkets also need to know the following:

(a) Market demand, which mainly includes the market demand for individual products;
(b) Consumer feedback and evaluation data;
(c) Supply chain data, including seasonal items, transportation time, delivery costs, etc.;
(d) Loss data, which includes items that may have been discounted but not sold.

In addition, other environmental factors can also be considered, such as the presence of adjacent supermarkets and the impact of online platforms, which can further improve the research in this paper.

*3.6. Advantages of the Model*

(1) The model does not add too many prior assumptions to the original data, nor does it add or delete too many factors to the original data after thorough data preprocessing and outlier detection and confirmation. This ensures the diversity and authenticity of the data, so that deeper features can be obtained from the data and the accuracy of subsequent tasks can be guaranteed.
(2) The model does not rely on artificial feature engineering, which ensures the transferability and robustness of the model.
(3) The model abandons the linear fitting scheme such as least squares. Instead, it seeks a fitting model based on machine learning, which can obtain better results in nonlinear cases.
(4) The model optimizes the backpack problem through the greedy strategy and obtains a quantitative category replenishment volume and pricing strategy, which appropriately fulfills the requirements.

**4. Conclusions**

In this paper, we implemented the automatic pricing and replenishment strategy of perishable products with a time-varying deterioration rate based on an improved SVR-LSTM-ARIMA hybrid model. We first calculated the ratio of cost-plus pricing through the sales price and cost, and then fit the total sales volume and cost-plus ratio using the least-squares method. Then, we calculated that there was no strong linear correlation between them, so we used the non-parametric support vector regression (SVR) method for fitting. Using data from the past 30 days to predict the situation in 7 days, a long short-term recurrent neural network and prior knowledge were used to extract time cycle information for weighted processing. The premise is to ensure the supply-and-demand relationship of supermarket products, calculate the sales volume of supermarkets, and use the trained SVR model to calculate the cost markup ratio to maximize supermarket profits. According to the correlation analysis, the linear correlation between the total sales volume and the markup ratio of pricing costs is relatively low. After examining the cost markup ratio of the total category, we found that, although there are a few outliers, the overall mean was 0.583, the sample standard deviation was 0.204, and the sample variance was small. In this article, we used the mean to fill in the missing values and then applied the SVR regression fitting algorithm to the data of each category to obtain a non-parametric relationship curve.

The fitting results generally fluctuated around the mean and were greatly influenced by time and other factors.

In order to predict the daily replenishment volume and pricing strategy of each vegetable category in the following week, we then examined the sales volume and cost markup ratio during this time period. The consumption of various categories was relatively stable in the past thirty days, and considering that 1–7 July 2023 is a complete week and there are no statutory holidays, working days and weekends also significantly affected sales. Although it is important to ensure maximum revenue for supermarkets, meeting sales remains the top priority. If one blindly pursues higher returns, which results in insufficient or excessive sales space, greater side effects occur, such as product expiration, insufficient warehouse storage capacity, and an inability to meet consumer demand, causing consumers to lose loyalty to the supermarket, which might lead to permanent loss of customer sources. Therefore, this article fits the sales data within the past 30 days using a long short memory recurrent neural network (LSTM) and uses the average MSE as the loss function of the model to directly predict the 7-day data to obtain the sales volume. To explore cycle information, the average sales volume within the past 30 days was directly used to calculate cycle weights, which were 0.175, 0.127, 0.110, 0.130, 0.143, 0.159, and 0.155, respectively. Linearly weighing the predicted results, we obtained the predicted sales volume. Then, we used regression fitting models to obtain pricing strategies. With the ARIMA model, past data can be used to make a simple prediction of the next day's price. Future values can be predicted based on the past and present values of the time series, attempting to predict the future unit price of goods solely based on the unit price data up to the current month. This article further optimized the grid strategy so that it can move the grid based on the predictions provided by ARIMA. In the model presented in this article, the movement of the grid is determined by the weighted sum of MA's long-term indicators and ARIMA's short-term indicators. By initializing the parameters in the grid movement formula provided in the article, we used the adjusted grid model to predict the price of the product.

Our improved SVR-LSTM-ARIMA hybrid model can also maintain the category of products as much as possible to meet customer needs, achieve mutual benefit, and provide a very reliable solution for solving practical problems. Our model can extract long-term, seasonal, cyclical, and irregular trends from data, which can be used for predicting economic development or trading in society, such as predicting the sales of cold drinks and clothing, as well as weather and temperature. Based on the problems encountered in prediction and analysis, practical experience, and the relevant literature in the field of economics, three suggestions for data collection are proposed from the perspective of pricing methods that balance cost, competition, and demand. First, to collect market average pricing and the pricing data from major competitors to balance economic benefits and competitive advantages. Second, to collect the expected price sensitivity of consumers to directly understand the satisfaction of the target consumer groups. Third, to understand the income and per capita consumption-level data of local residents, and to understand the overall consumption ability and willingness of residents. Our model can also maintain the category of products as much as possible to meet customer needs, achieve mutual benefit, and provide a very reliable solution for solving practical problems.

**Author Contributions:** Writing—original draft preparation, A.G.; writing—review and editing, A.G.; data curation, Z.Y., X.Z. and Y.X.; software, Z.Y., X.Z. and Y.X.; funding acquisition, A.G. All authors have read and agreed to the published version of the manuscript.

**Funding:** This work is supported by Project of Industry, School and Research Institutions in Jiangsu Province of China under Grant no. BY20230000.

**Data Availability Statement:** The real datasets used were obtained from the 2023 Higher Education Society Cup National College Student Mathematical Modeling Competition Questions of China at http://www.mcm.edu.cn/html_cn/node/c74d72127066f510a5723a94b5323a26.html (accessed on 7 September 2023).

**Conflicts of Interest:** The authors declare no conflict of interest.

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
