# Peer review of "Research on the Modeling of Automatic Pricing and Replenishment Strategies for Perishable Goods with Time-Varying Deterioration Rates"

_axioms, doi:10.3390/axioms13010062_

Round 1

Reviewer 1 Report

Comments and Suggestions for Authors

The goal of this study is to develop automatic pricing and replenishment strategies for perishable products with time-varying deterioration rates. To this end, SVR and LSTM models were considered. Although the topic is interesting, authors need to consider the following comment to revise the manuscript.   

In Line 98: Authors need to explain how they collected data. If there was actual experiment to collect data, authors should make a new section to explain about their data collection effort. 

In Line 99: All symbols used in equations should be clearly defined. Also, references are missing. It is clear that the SVR model is not provided by authors of this paper. In fact, regarding that authors did not cite relevant articles to support their claim, this reviewer is wondering whether the authors conducted rigorous literature review before they conduct this study.

In Line 109: Similar to SVR, many relevant references are not cited. 

In Line 143: ARIMA is also existing method. Please clearly address authors contribution in the methodology section. It seems like authors just used three existing algorithms (i.e., SVR, LSTM, and ARIMA) to develop the pricing and replenishment strategies. It will be a good idea to develop an overall framework using SVR, LSTM, and ARIMA to solve the research problem. 

In Line 156: All figures and tables should be explained and analyzed. It seems like authors just illustrate how they utilize the existing approach for processing of commodity and sales data. Please show academic contribution of this study as a scientific article.   

In Line 450: Conclusion is too short. Similar to the abstract, it should summarize the study and address its limitations (or future work)

There are multiple formatting issues as an academic papers. Please check the template (or sample papers) provided by MDPI.

In Line 68: "the support vector regression SVR non-parametric method" should be written as "the support vector regression (SVR) non-parametric method"

In Line 107: The format of Equations (1) and (2) are different from the MDPI template. (please update other equations as well)

In Line 459: Please check the reference format.

Author Response

(I) Please check that all references are relevant to the contents of the manuscript.

Responses:  Thanks for this comment. We've added some references and they can be reflected in the article.

Reviewer 2 Report

Comments and Suggestions for Authors

The paper focuses on developing automated pricing and replenishment strategies for perishable goods, specifically examining the vegetable category in a supermarket setting in China. It aims to maximize revenue by analyzing historical sales data, applying various models like Support Vector Regression (SVR), Long Short-Term Memory (LSTM), and Autoregressive Integrated Moving Average (ARIMA) to predict sales volumes and cost-plus pricing. The paper also discusses data preprocessing methods, outlier detection, and filtering techniques. Additionally, it proposes a greedy algorithm to optimize the replenishment plan for single products based on maximizing revenue while meeting market demand and constraints.

Strengths:

1) Comprehensive Approach: The paper covers a wide array of analytical methods (SVR, LSTM, ARIMA) for modeling sales volume and pricing strategies, showcasing a comprehensive approach to address the problem.

2) Detailed Data Preprocessing: The paper provides a thorough explanation of data preprocessing steps, including outlier detection, missing value imputation (using KNNImputer), and data cleaning methods like Boxplot analysis, ensuring data reliability.

3) Model Diversity: Utilizing machine learning models (SVR, LSTM) instead of conventional linear models and incorporating different techniques for forecasting and optimization enhances model performance and flexibility.

4) Application of Greedy Algorithm: The utilization of the greedy algorithm for optimizing the replenishment plan for single products demonstrates an attempt to solve a complex problem in a practical setting.

Weaknesses:

1) The Abstract is too long and needs to be shortened and more focused.

2) Lack of Clarity in Explanations: Some sections lack clarity in explanations, making it challenging to understand the methodology and processes. This can hinder reproducibility and understanding for readers unfamiliar with the techniques used.

3) Insufficient Evaluation Metrics: The paper lacks detailed evaluation metrics or comparison with alternative models, making it difficult to assess the effectiveness and superiority of the proposed models over traditional methods.

4) Incomplete Consideration of Factors: The paper mainly focuses on historical sales data and models, overlooking other potentially influential factors such as external market dynamics, consumer behavior changes, or competitor strategies, which could impact the model's accuracy.

5) Limited Validation and Real-world Implementation: The proposed models lack validation with separate test data or real-world implementation results, which are crucial for verifying the models' effectiveness in practical scenarios.

6) The Conclusions section is too short and completely lacks a detailed discussed of the merits and the limits of your analysis.

Comments on the Quality of English Language

.

Author Response

(II)  Any revisions to the manuscript should be highlighted, such that any changes can be easily reviewed by editors and reviewers.

Responses: Thank you for your kind comment, we have highlighted it in the article, I believe readers will also have an easy read.

Reviewer 3 Report

Comments and Suggestions for Authors

Comments on the Quality of English Language

The English language needs major editing.

Author Response

(III) Please provide a cover letter to explain, point by point, the details of the revisions to the manuscript and your responses to the referees’ comments.

Responses:  Thank you for your comment, we have revised and responded to each detail.

Reviewer 4 Report

Comments and Suggestions for Authors

Dear authors,

Please see attached my review.

Comments on the Quality of English Language

Author Response

(IV) If you found it impossible to address certain comments in the review reports, please include an explanation in your appeal.

Responses:  Thanks for your comment, we've all managed to solve it.

Reviewer 5 Report

Comments and Suggestions for Authors

The review is attached.

Author Response

(V) The revised version will be sent to the editors and reviewers.

Responses:  Ok, thank you for your valuable comments .

Round 2

Reviewer 1 Report

Comments and Suggestions for Authors

Most of major comments have been appropriately incorporated in the manuscript. But, it is still have some minor formatting issue.

In Line 107: In general, numbers of equations are  aligned with a right edge. Please check the equation format of the MDPI template. (please resolve the alignment issue for other equations as well)

Author Response

The authors would like to express their appreciation to the reviewer for his/her comments and the efforts provided in helping us to improve the quality and presentation of the paper further.

As for the specific comments made, we have the following responses:

(1)In Line 107: In general, numbers of equations are  aligned with a right edge. Please check the equation format of the MDPI template. (please resolve the alignment issue for other equations as well)

Responses:Thank you for your valuable comments. We have modified all formula formats of the MDPI template.

Reviewer 2 Report

Comments and Suggestions for Authors

Nothing to add.

Comments on the Quality of English Language

.

Author Response

The authors would like to express their appreciation to the reviewer for his/her comments and the efforts provided in helping us to improve the quality and presentation of the paper further.

As for the specific comments made, we have the following responses:

We have made significant changes to the introduction, conclusion, and references. We have checked our paper: there is inconsistency in the terminology before and after; Machine translation comprehension error; Grammar errors; No subject sentence; Formula omission; Mistranslation. We have reviewed every sentence to ensure there are no grammar errors.

Reviewer 3 Report

Comments and Suggestions for Authors

I cannot see no essential change in the revised manuscript which to makes me change my previous opinion. Thus I think that the paper has to be rejected.

Comments on the Quality of English Language

The English language needs additional minor editing.

Author Response

(The authors gave the same response as above.)

Round 3

Reviewer 3 Report

Comments and Suggestions for Authors

I still cannot see enough changes in the revised manuscript. For example, there are still many  figures and tables which are not discussed.  Thus I think that the paper has to be rejected.